# NOISE-BASED REGULARIZERS FOR RECURRENT NEURAL NETWORKS

## ABSTRACT

Recurrent neural networks (RNNs) are powerful models for sequential data. They can approximate arbitrary computations, and have been used successfully in domains such as text and speech. However, the flexibility of RNNs makes them susceptible to overfitting and regularization is important. We develop a noise-based regularization method for RNNs. The idea is simple and easy to implement: we inject noise in the hidden units of the RNN and then maximize the original RNN's likelihood averaged over the injected noise. On a language modeling benchmark, our method achieves better performance than its deterministic RNN and variational dropout (Gal and Ghahramani, 2016).

## 1 INTRODUCTION

Recurrent neural networks (RNNs) are a class of powerful and flexible deep sequential models (Robinson and Fallside, 1987; Werbos, 1988; Williams, 1989; Elman, 1990; Pearlmutter, 1995). They compute probabilities over sequences by conditioning on a low dimensional hidden state that represents a summary of the past sequence. RNNs enjoy universal approximation capabilities and are Turing complete (Siegelmann and Sontag, 1991; Hammer, 2000). They have achieved state-of-the-art results on many applications (Mikolov and Zweig, 2012; Graves, 2013; Gregor et al., 2015). RNNs have benefited from architectural improvements (Hochreiter and Schmidhuber, 1997; Cho et al., 2014) and improved optimization (Martens and Sutskever, 2011; Bengio et al., 2013).

The success of RNNs in many domains stems from their ability to approximate arbitrary computations. However, the flexibility of RNNs also creates a challenge. RNNs easily memorize observations; they overfit. Regularizing them prevents overfitting. Many techniques have been developed to address this overfitting issue for RNNs. These techniques focus on limiting the form of the weight matrix used in the RNN's recurrent dynamics. They prune by dropping updates to the state of the RNN (Krueger et al., 2016; Semeniuta et al., 2016), and they also directly limit the form of the weight matrix (Cooijmans et al., 2016). More recently Merity et al. (2017) have explored and extended these regularization techniques for state-of-the-art results on language modeling.

In this paper, we develop a new way to regularize RNNs called NOISIN. The idea is simple. NOISIN works by injecting noise into the RNN's transition dynamics such that the expected value of each state transition matches the RNN's transition. The corresponding objective function takes this transition and averages over the injected noise. Figure 2 demonstrates the value of this kind of regularization. Traditional optimization for the RNN quickly overfits; when using NOISIN both the training loss and held-out loss continue to improve.

The added noise in the objective regularizes by smoothing out the optimization landscape. This smoothing blurs together parts of the original objective by averaging over the neighborhood of the local optima. Because of this averaging, the objective forms a lower bound on the log likelihood of the data. We develop an algorithm that uses the same gradient information as the RNN, thereby regularizing with little added computational cost. Our algorithm optimizes the regularization parameter, the spread of the added noise. This optimization is theoretically grounded in empirical Bayes (Robbins, 1964). The overall technique is simple and can be applied to any recurrent neural network architecture. We demonstrate the effectiveness of NOISIN on language models. Regularizing with noise helps the RNN learn with less fear of overfitting.

**Related work** : A lot of attention has been given to finding efficient ways to regularize RNNs. We classify each of these works into one of the following categories.

Weight-based regularization: One style of weight-based regularization comes from adding noise to the weights of the RNN. The most successful method in this category is dropout (Srivastava et al., 2014; Gal and Ghahramani, 2016) and its variants (Zaremba et al., 2014; Wan et al., 2013). The other broad style of weight-based regularization put constraints on the form of the recurrent weight matrix such as orthogonality (Wisdom et al., 2016; Arjovsky et al., 2016). These approaches tend to be computationally expensive in that they require multiple representations of the weight matrix or optimization over complicated manifolds.

Activation-based regularization: these methods introduce auxiliary parameters and apply some form of normalization (Ioffe and Szegedy, 2015; Ba et al., 2016; Cooijmans et al., 2016). Though powerful, these approaches require keeping track of more statistics that consumes memory that could be used to enhance the underlying model.

Hidden unit-based regularization: these methods reduce model capacity by randomly skipping the updates for some hidden units (Krueger et al., 2016; Semeniuta et al., 2016). NOISIN falls into this category. Instead of skipping updates, NOISIN injects noise directly into the hidden units to regularize the RNN. The objective we define is the log likelihood averaged over the noise distribution. This style of objective has been used in different domains to transform discrete optimization problems into smooth optimization problems (Xu et al., 2015; Louppe and Cranmer, 2017). We focus on a continuous setting to regularize RNNs.

## 2 BACKGROUND

In this section we briefly review the key ingredients used in our method: recurrent neural networks and exponential family distributions.

### 2.1 RECURRENT NEURAL NETWORKS

Recurrent neural networks (RNNs) are models for sequences. The joint distribution of a sequence of observations $(\mathbf{x}_1, ..., \mathbf{x}_T)$, can be factorized according to the chain rule of probability:

$$p(\mathbf{x}_1, ..., \mathbf{x}_T) = p(\mathbf{x}_1) \prod_{t=2}^{T} p(\mathbf{x}_t | \mathbf{x}_{1:t-1}).$$

RNNs compute each conditional probability $p(\mathbf{x}_t | \mathbf{x}_{1:t-1})$ via a low dimensional recurrent hidden state that represents a summary of the sequence up to time $t$,

$$\mathbf{h}_t = f(\mathbf{x}_{t-1}, \mathbf{h}_{t-1}; U, V, W) \text{ and } p(\mathbf{x}_t | \mathbf{x}_{1:t-1}) = p(\mathbf{x}_t | \mathbf{h}_t).$$

Here the function $f$ describes the transition mechanism for the recurrent network (one example of such transition mechanism is gating (Hochreiter and Schmidhuber, 1997; Cho et al., 2014)). The weights $U, W$, and $V$ are parameters of the network. We dropped the biases for simplicity.

RNNs are trained via backpropagation through time. Backpropagation through time builds gradients by unrolling the recurrent neural network into a feed-forward neural network and applies backpropagation (Rumelhart et al., 1988) on the feed-forward neural network. The RNN is then optimized using gradient descent or stochastic gradient descent (Robbins and Monro, 1951). This yields a simple procedure for training RNNs.

### 2.2 EXPONENTIAL FAMILIES

The exponential family (Brown, 1986) is a broad class of distributions. Almost all of the distributions used in practice are members of the exponential family: Gaussian, Gumbel, Beta, Gamma, Poisson, Bernoulli, Exponential, Categorical, and Dirichlet to name a few. This generality contributes to both convenience and larger scale understanding. A probability distribution in the exponential family has

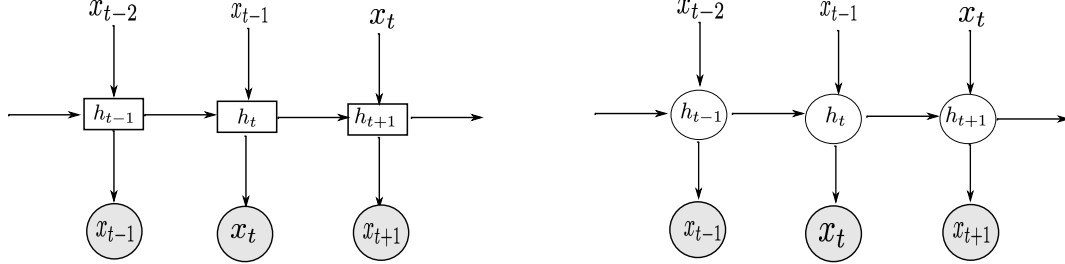

**Figure 1: left**: a deterministic one-layer RNN whose hidden state at time step $t$ is defined as $\mathbf{h}_t = f(\mathbf{x}_{t-1}, \mathbf{h}_{t-1})$. **right**: a noisy one-layer RNN with latent hidden state at time step $t$ defined as $\mathbf{h}_t \sim \text{ExpFam}(\mathbf{h}_t; \eta(\mathbf{x}_{t-1}, \mathbf{h}_{t-1}))$. Choosing one exponential family distribution over another amounts to changing the nature of the injected noise and thus the regularization procedure for the RNN.

the following canonical form:

$$p(\mathbf{x}) = \exp\left(\eta^T t(\mathbf{x}) - A(\eta)\right)$$

$$A(\eta) = \log \int \exp\left(\eta^T t(\mathbf{x})\right) \nu(d\mathbf{x})$$

where $A(\eta)$ is called the log partition function (or log normalizer), $\eta$ is called the natural parameter, and $t(\mathbf{x})$ denotes the vector of sufficient statistics. In what follows, we inject exponential family distributed noise into the RNN to perform regularization.

## 3 NOISE-NASED REGULARIZERS FOR RNNS

In this section we first motivate the value of noise in RNNs and discuss desiderata for the noise distribution. Then we introduce NOISIN, a noise-based regularization method for RNNs. We finally connect our method to two main theoretical ideas: variational inference and empirical Bayes.

**Motivating noise injection in RNNs:**    Adding noise to RNNs enables broader exploration of the network's parameter space. It can be done at different levels in the RNN's hierarchy. We focus on the input level and the recurrent level.

At the input level this is dataset augmentation (Goodfellow et al., 2016). Each input is perturbed with some noise before being conditioned on in the recurrent layer. This approach helps in settings where the observations are high dimensional such as in object recognition, language modeling, and speech recognition (Jaitly and Hinton, 2013; Goodfellow et al., 2016). Note that it is in these high dimensional settings that overfitting frequently occurs: the high dimensionality of the inputs often result in a very large number of parameters for the RNN (the input-to-hidden and hidden-to-output matrices in RNNs scale with the dimensionality of the data).

For the weights, noise injection can be helpful. This type of noise can be cast in the Bayesian paradigm. It has been successfully applied to reduce the complexity of RNNs (Gal and Ghahramani, 2016). Alternatively, noise can be injected into the hidden states. We develop this approach here.

**Desiderata for noise:**    Noise can be used to regularize RNNs, but injecting noise in an arbitrary manner can lose the meaning of the underlying model. Therefore the noise needs to meet certain requirements to preserve the original RNN. Denote by $\mathbf{z}_t$ the noisy version of the deterministic hidden state $\mathbf{h}_t$ at time step $t$. There are three requirements for injecting noise in RNNs:

1. The noisy hidden state $\mathbf{z}_t$ should have expectation $\mathbf{h}_t$ conditional on the previous noisy hidden state. $E(\mathbf{z}_t|\mathbf{z}_{t-1}) = \mathbf{h}_t$,

2. The noise should have a free parameter $\gamma$ controlling its entropy or spread,

3. Backpropagating through the log likelihood of the data should still include the gradients from the original RNN.

The first desideratum puts constraints on the first moment of the noise. It is to make sure the underlying architecture is preserved.

The second condition puts a constraint on the spread of the noise (like the variance). This control on the noise level determines the amount of regularization one wants to use. It is like controlling the amount of regularization in any machine learning model. For example if the noise is Gaussian, the free parameter for the noise should be the variance. If the noise is Gamma, the parameter should be the scale or rate parameter of the Gamma normalized by the shape. This regularization parameter $\gamma$ can then be tuned via hyperparameter search using a validation set or it can be determined by the data itself.

**Noisy RNNs** We now describe the regularization method for RNNs based on exponential families. Assume a sequence of observations $\mathbf{x}_1, ..., \mathbf{x}_T$. We inject noise in the hidden units of a one-layer RNN via the following generative process:

1. Draw $\mathbf{z}_t \sim ExpFam\Big(\mathbf{z}_t; f(U\mathbf{x}_{t-1} + W_L\mathbf{z}_{t-1}), \gamma\Big)$
2. Draw $\mathbf{x}_t \sim p(\mathbf{x}_t|\mathbf{h}_t = \mathbf{z}_t)$

This generative process defines a recurrent neural network with noisy hidden states. The noise can be additive or multiplicative depending on the chosen exponential family distribution. At each time step in the sequence, the hidden state $\mathbf{z}_t$ is modeled as an exponential family whose mean and spread are $f(U\mathbf{x}_{t-1} + W_L\mathbf{z}_{t-1})$ and $\gamma$ respectively. The spread $\gamma$ is shared by the units in the layer. It controls the noise level injected in the hidden units and can be thought of as a regularization parameter. Learning $\gamma$ is equivalent to learning a regularization hyperparameter. This can be done via hyperparameter search.

As an example, consider injecting a Gaussian noise into the hidden states. Further assume the observations are categorical as in language modeling. Then the generative process becomes:

1. $\mathbf{z}_t = f(U\mathbf{x}_{t-1} + W_L\mathbf{z}_{t-1}) + \epsilon \cdot \gamma$ where $\epsilon \sim \mathcal{N}(0, \mathbf{I})$
2. $\mathbf{x}_t \sim p(\mathbf{x}_t|\mathbf{h}_t = \mathbf{z}_t) = \text{softmax}(V\mathbf{z}_t)$

Notice we recover the deterministic RNN by setting $\gamma$ to zero.

The construction above meets the noise desiderata. The conditional expectation of the latent hidden state conditional on the last hidden state and the previous observation is the original RNN,

$$E(\mathbf{z}_t|\mathbf{z}_{t-1}) = f(U\mathbf{x}_{t-1} + W\mathbf{z}_{t-1}).$$

Extending this construction to multilayer RNNs is straightforward:

1. Draw $\mathbf{z}_t^L \sim ExpFam\Big(\mathbf{z}_t^L; f(U\mathbf{x}_{t-1} + W_L\mathbf{z}_{t-1}^L), \gamma_L\Big)$
2. For $l$ from $L - 1$ to 1:
   (a) Draw $\mathbf{z}_t^l \sim ExpFam\Big(\mathbf{z}_t^l; f(W_l\mathbf{z}_{t-1}^l + Q_l\mathbf{z}_t^{l+1}), \gamma_l\Big)$
3. Draw $\mathbf{x}_t \sim p(\mathbf{x}_t|\mathbf{h}_t = \mathbf{z}_t)$

This defines a deep recurrent neural network with $L$ layers and noisy hidden states. At each time step in the sequence, the hidden state $\mathbf{z}_t^l$ at layer $l$ is modeled as an exponential family whose mean follows the transition dynamics of the RNN. The spread $\gamma_l$ is shared by the units in the same layer but not across layers.

To meet the noise desiderata, we enforce a constraint on the expectation of the hidden state,

$$E(\mathbf{z}_t^L|\mathbf{z}_{t-1}^L) = f(U\mathbf{x}_{t-1} + W_L\mathbf{z}_{t-1}^L) \text{ and } E(\mathbf{z}_t^l|\mathbf{z}_{t-1}^l, \mathbf{z}_t^{l+1}) = f(W_l\mathbf{z}_{t-1}^l + Q_l\mathbf{z}_t^{l+1}).$$

This constraint recovers the original RNN when the noise variance is zero.

**Table 1:** Summary statistics of the datasets used for the experiments.

| | Penn Treebank | | | WikiText-2 | | |
| | Train | Valid | Test | Train | Valid | Test |
| --- | --- | --- | --- | --- | --- | --- |
| Tokens | 929,590 | 73,761 | 82,431 | 2,088,628 | 217,646 | 245,569 |
| Vocab Size | | 10,000 | | | 33,278 | |
| OoV Rate | | 4.8% | | | 2.6% | |

Although we have illustrated our method using the simplest RNN, it can be applied to other architectures such as LSTMs or GRUs by varying the transition function. We study both the RNN and LSTM in our experiments.

**Learning procedure**  Consider the one-layer noisy RNN. To learn the model parameters we maximize the expected log likelihood under the implied prior on the latent hidden states:

$$\mathcal{L} = E_{p(\mathbf{z}_{1:T})} \Big[ \log p(\mathbf{x}_{1:T} | \mathbf{z}_{1:T}) \Big] \tag{1}$$

This loss is a lower bound on the log marginal likelihood of the data:

$$\mathcal{L} = E_{p(\mathbf{z}_{1:T})} \Big[ \log p(\mathbf{x}_{1:T} | \mathbf{z}_{1:T}) \Big] \leq \log \Big[ E_{p(\mathbf{z}_{1:T})} p(\mathbf{x}_{1:T} | \mathbf{z}_{1:T}) \Big] = \log p(\mathbf{x}_{1:T})$$

where we used the concavity of the logarithm and Jensen's inequality. Consider the structured factorization of the distribution on the hidden states. We rewrite the loss function as a sum over the sequence length,

$$\mathcal{L} = \sum_{t=1}^{T} \mathcal{L}_t \text{ and } \mathcal{L}_t = E_{p^*(\mathbf{z}_{t-1})} E_{p(\mathbf{z}_t | \mathbf{z}_{t-1}, \mathbf{x}_{t-1})} \Big[ \log p(\mathbf{x}_t | \mathbf{z}_t) \Big].$$

Here $p^*(\mathbf{z}_{t-1})$ denotes the marginal distribution of the past hidden state. The objective corresponds to averaging the predictions of infinitely many RNNs at each time step in the sequence. This is known as an ensemble method and has a regularization effect. However ensemble methods are costly as it requires training all the sub-models in the ensemble. With our approach, at each time step in the sequence, one of the infinitely many RNNs is trained and because of the parameter sharing, the RNN being trained at the next time step will use better settings of the weights. This makes training the whole network efficient.

NOISIN approximates the loss with Monte Carlo,

$$\mathcal{L}_t = \frac{1}{S} \sum_{s=1}^{S} \log p(\mathbf{x}_t^s | \mathbf{z}_t^s).$$

The hidden states are sampled via ancestral sampling,

$$\mathbf{z}_{t-1}^s \sim p^*(\mathbf{z}_{t-1}^s) \text{ and } \mathbf{z}_t^s \sim p(\mathbf{z}_t^s | \mathbf{z}_{t-1}^s, \mathbf{x}_{t-1}).$$

Because of the restrictions on the noise, the objective above can be optimized using noisy gradient steps and stochastic gradient descent. The training procedure is as easy as for the deterministic RNN when using one sample during training and averaging at test time.

One way we tune the noise level hyperparameter in Section 4 is to use gradient updates and learn it along with the other model parameters. Care needs to be taken in doing this. If we put no constraint during training on the noise level, the algorithm drives this parameter to zero, making the noisy RNN become deterministic. This makes the initial noise injection useless. However there are two easy solutions to this issue: either put a lower bound on the variance or use a validation set as a means to stop the training procedure. We experiment with this approach and grid search fo the hyperparameter in Section 4.

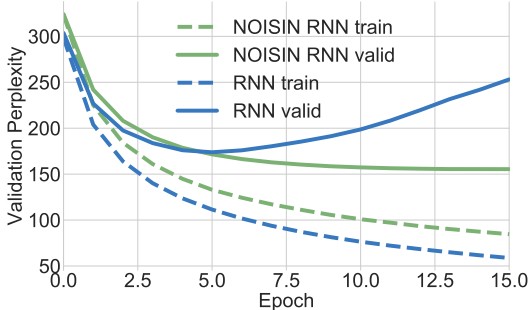

**Figure 2: Left**: validation perplexity for the deterministic RNN and the RNN with NOISIN regularization. **Right**: the corresponding training perplexity. The settings were the same in both cases. We used the sigmoid activation function on a one-layer recurrent neural network with 256 hidden units. The RNN overfits after only five epochs, and its training loss still decreases. This is not the case for the RNN with NOISIN regularization.

**Connections and extensions.**    We illustrate two connections with our method.

- **NOISIN as variational inference**: consider the same set up as above where we have observations $\mathbf{x}$ and unobserved latent variables $\mathbf{z}$. Variational inference is a family of algorithms for learning the distribution $p(\mathbf{z}|\mathbf{x})$. It does so by positing a parameterized approximating distribution $q(\mathbf{z}; \phi)$ and optimizing a divergence $KL(q(\mathbf{z}; \phi)||p(\mathbf{z}|\mathbf{x}))$ between the true posterior distribution and the approximate posterior. This objective is intractable but a lower bound on the log likelihood known as the ELBO can be used as a proxy for learning:

$$\text{ELBO} = E_{q(\mathbf{z};\phi)}\Big[\log p(\mathbf{x}|\mathbf{z})\Big] - KL\Big[q(\mathbf{z};\phi)||p(\mathbf{z};\psi)\Big].$$

Assume $p(\mathbf{z}; \psi) = q(\mathbf{z}; \phi)$ a.e. With this choice of variational distribution, the KL term above is zero. The ELBO then becomes the Noisin' objective:

$$\text{ELBO} = E_{p(\mathbf{z};\psi)}\Big[\log p(\mathbf{x}|\mathbf{z})\Big].$$

The ELBO, in the context of deep learning, has been widely used to learn neural network parameters using the Bayesian paradigm. However successfully optimizing the ELBO often requires ad-hoc techniques to prevent the KL term from vanishing. Instead of adding extra sets of parameters (the variational parameters) and optimizing these parameters to explain the data well while staying close to the prior, we optimize an objective that uses an unknown prior whose parameters are learned with data.

- **NOISIN and empirical Bayes**: consider a one-layer noisy RNN with latent weights. It's joint distribution is:

$$p(\mathbf{x}_{1:T}, \mathbf{z}_{1:T}) = p(\mathbf{x}_{1:T}|\mathbf{z}_{1:T}) \cdot p(\mathbf{z}_{1:T}|W, U) \cdot p(W; \eta_w) \cdot p(U; \eta_u)$$

This is a hierarchical model. Empirical Bayes consists in getting point estimates of the hyperparameters $\eta_w$ and $\eta_u$ using data then using those point estimates to learn the posterior distribution of the hidden states and weights. When the weights are deterministic this is equivalent to putting Dirac priors on them. In this case learning the hyperparameters boils down to learning the weights. When we optimize the objective in Equation 1, we are implicitly learning the priors on the weights and conditioning on these weights to learn the approximate posterior distribution of the hidden states. This approximate posterior distribution in our case is the approximation mentioned above when discussing variational inference.

## 4    EMPIRICAL STUDY

We assess the effectiveness of the regularization technique introduced in this paper against the deterministic RNN and variational dropout (Gal and Ghahramani, 2016) on language modeling. We describe below the data, the experimental settings, and the results.

**Table 2:** NOISIN improves generalization performance in recurrent neural network language models. The results below correspond to perplexity scores on the test set of the Penn Treebank. The lower the perplexity the better. We compare to the version of dropout implemented in (Gal and Ghahramani, 2016). An asterisk next to NOISIN results indicates that the variance parameter of the Gaussian noise distribution was learned. Overall we find that learned noise had more stable performance than fixed noise. The computational cost of NOISIN and the traditional RNN were similar, while dropout was noticeably slower.

| Method | Layers | Activation | Perplexity |
|--------|--------|------------|------------|
| NOISIN | 1 | tanh | **131.9** |
| NOISIN* | 1 | tanh | 140.9 |
| Dropout | 1 | tanh | 138.3 |
| None | 1 | tanh | 141.4 |
| NOISIN | 2 | tanh | **140.5** |
| NOISIN* | 2 | tanh | 142.4 |
| Dropout | 2 | tanh | 165.2 |
| None | 2 | tanh | 157.9 |
| NOISIN | 1 | sigmoid | 166.0 |
| NOISIN* | 1 | sigmoid | **144.7** |
| Dropout | 1 | sigmoid | 146.9 |
| None | 1 | sigmoid | 157.4 |
| NOISIN | 2 | sigmoid | 196.8 |
| NOISIN* | 2 | sigmoid | 154.3 |
| Dropout | 2 | sigmoid | **152.2** |
| None | 2 | sigmoid | 164.4 |
| NOISIN | 1 | rectifier | 158.4 |
| NOISIN* | 1 | rectifier | **149.2** |
| Dropout | 1 | rectifier | 169.8 |
| None | 1 | rectifier | 175.3 |
| NOISIN | 2 | rectifier | 171.0 |
| NOISIN* | 2 | rectifier | 165.9 |
| Dropout | 2 | rectifier | 168.5 |
| None | 2 | rectifier | **165.0** |

**Data:** We use two datasets: the Penn Treebank benchmark for language modeling (Marcus et al., 1993) and the recently introduced Wikitext-2 dataset (Merity et al., 2016). Table 1 summarizes the characteristics of these datasets.

**Experimental settings:** The models were implemented in MXNet (Chen et al., 2015). For the noisy RNN models we chose Gaussian noise for simplicity. Any of the existing exponential family distributions can be used for the noise distribution. We found initialization to play a very big role for both the noisy RNN and the deterministic RNN. We provide more details on the experimental settings for each dataset below.

**Results on the Penn Treebank.** We used the Adam (Kingma and Ba, 2014) algorithm with a fixed learning rate of 0.001. We report the perplexity results on Table 2. The regularized RNNs overall perform better than the deterministic RNN. NOISIN yields better perplexity scores than the deterministic RNN and variational dropout on all but one instance. We attribute this performance to the fact that the hidden layers are fully stochastic. Adding stochasticity to the hidden layers and keeping weights deterministic yields better generalization capabilities (Chung et al., 2015; Fraccaro et al., 2016; Bayer and Osendorfer, 2014; Fortunato et al., 2017).

**Table 3:** Test set perplexity scores on the Wikitext-2 dataset. The lower the better. NOISIN leads to better performance.

| Method | Layers | Size | Perplexity |
|--------|--------|------|------------|
| NOISIN | 2 | 200 | **104.7** |
| None | 2 | 200 | 105.8 |

**Results on the Wikitext-2 dataset.** We followed the setup in Zaremba et al. (2014). We used a two-layer LSTM with 200 hidden units in each layer. We used stochastic gradient descent with an initial learning rate of 1 and a minibatch size of 32 for computational efficiency. We clipped the global norm of the gradients to a maximum of 5. After four epochs of training, we divided the learning rate by 2 every epoch. For NOISIN with Gaussian noise, we learned the standard deviation parameter. To avoid driving the regularization parameter to zero, we added a constant of 0.1. This is a lower bound on the variance and ensures the noisy RNN does not behave identically as the deterministic RNN. This improved performance. We report the perplexity results on Table 3. For computational reasons we only compare to the deterministic RNN. The variational dropout takes longer to train and converges slower.

The difference with the deterministic RNN in this case is minor. We attribute this to the following: the wikitext-2 dataset is relatively large. The size of the network we chose is relatively small. Choosing a bigger number of hidden units would improve performance for both models but because of overfitting, the deterministic RNN would overfit more quickly than any regularization method.

## 5 DISCUSSION

We proposed a simple and easy to implement method for regularizing RNNs that consists in injecting an exponential family distributed noise into the hidden units. The resulting network is learned by maximizing a lower bound to the log marginal likelihood of the data – the expected log likelihood under the hidden states prior. This objective is theoretically grounded in empirical Bayes and has connections to variational inference. Its regularization capabilities stem from an averaging of the predictions of an infinite number of RNNs. This ensemble method interpretation of the objective is also a characteristic of dropout – a very successful regularization technique for neural networks. We showed in a language modeling benchmark that the proposed method yields better generalization capabilities than its deterministic counterpart and the variational dropout.

How can we go from here? This noise injection framework opens the rich literature about exponential families to regularization in deep sequential models. The main issue in sequential modeling is capturing long-term dependencies. The natural question that arises then is whether there is a distribution in this class of exponential family distributions that is the right noise distribution to capture long-term dependencies. And if so, which one is it? If not, what are the desired properties of such a distribution and how can we derive it? We leave this as future work.

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
