# OpenReview forum: "Noise-Based Regularizers for Recurrent Neural Networks"
_ICLR.cc/2018/Conference — Reject_

### Official Review · AnonReviewer1 · 2017-11-27
**Sever issues with prior work and justification**

**Rating:** 2
**Confidence:** 5

**Review:**

The authors of the paper advocate injecting noise into the activations of recurrent networks for regularisation. This is done  by replacing the deterministic units with stochastic ones.

The paper has several issues with respect to the method and related work.

- The paper needs to mention [Graves 2011], which is one of the first works to inject noise into the dynamics of an RNN. It is also important to know how these two approaches differ. E.g.: Under what conditions are the two approaches equivalent? How do they compare experimentally?
- While [Bayer & Osendorfer, 2014] and [Chung et al, 2015] appear in the list of references, these works are not discussed in the main text. I personally think these are extremely related, pioneering the use of stochastic units in a recurrent context. In the end, the original paper can be cast in these frameworks approximately by removing the KL term of the ELBO. This might be ok by itself, but that the authors are apparently aware of the work (as it is in the list of references) and not discussing them in the main text makes me highly skeptical.
- The method is introduced for general exponential families, but a) not empirically evaluated for more than the Gaussian case and b) not a complete algorithm for e.g. the Bernoulli case. More specifically, the reader is left alone with the problem of estimating the gradients in the Bernoulli case, which is an active area of research by itself.
- The paper makes use of the reparameterisation trick, but does not cite the relevant literature, e.g. [Kingma 2013, Rezende 2014, and another one I currently struggle to find].
- The desiderate for noise seem completely arbitrary to me and are not justified. I don’t see why violation of any of them would lead to an inferior regularisation method.

### References
[Graves 2011] Graves, Alex. "Practical variational inference for neural networks." Advances in Neural Information Processing Systems. 2011.
[Kingma 2013] Kingma, Diederik P., and Max Welling. "Auto-encoding variational bayes." arXiv preprint arXiv:1312.6114 (2013).
[Rezende 2014] Rezende, Danilo Jimenez, Shakir Mohamed, and Daan Wierstra. "Stochastic backpropagation and approximate inference in deep generative models." arXiv preprint arXiv:1401.4082 (2014).

---

### Official Review · AnonReviewer3 · 2017-11-27
**Sample hidden states of an RNN instead of predicting them deterministically. Interesting idea that is insufficiently explored.**

**Rating:** 5
**Confidence:** 4

**Review:**

The RNN transition function is: h_t+1 = f(h_t,x_t)
This paper proposes using a stochastic transition function instead of a deterministic one.
i.e h_{t+1} \sim expfam(mean = f(h_t,x_t), gamma) where expfam denotes a distribution from the exponential family.

The experimental results consider text modeling (evaluating on perplexity) on Penn Treebank and Wikitext-2. The method of regularization is compared to a reimplementation of Variational Dropout and no regularization.

The work is written clearly and easy to follow.

Overall, the core idea in this work is interesting but underexplored.

* As of when I read this paper, all results on this work used 200 hidden units realizing results that were well off from the state of the art results on Penn Tree Bank (as pointed out by the external reader).
The authors responded by stating that this was done to achieve a relative comparison.  A more interesting comparison, in addition to the ones presented, would be to see how well each method performs while not controlling for hidden layer size. Then, it might be that restricting the number of hidden dimensions is required for the RNN without any regularization but for both Variational Dropout and Noisin, one obtains better results with a larger the hidden dimension.

* The current experimental setup makes it difficult to assess when the proposed regularization is useful. Table 2 suggests the answer is sometimes and Table 3 suggests its marginally useful when the RNN size is restricted.

* How does the proposed method's peformance compare to Zoneout https://arxiv.org/pdf/1606.01305.pdf?

* Clarifying the role of variational inference: I could be missing something but I don't see a good reason why the prior (even if learned) should be close to the true posterior under the model. I fear the bound in Section (3) [please include equation numbers in the paper] could be quite loose.

* What is the rationale for not comparing to the model proposed in [Chung et. al] where there is a stochastic and deterministic component to the transition function? In what situations do we expect the fully stochastic transition here to work better than a model that has both? Presumably, some aspect of the latent variable + RNN model could be expressed by having a small variance for a subset of the dimensions and large one for the others
but since gamma is the same across all dimensions of the model, I'm not sure this feature can be incorporated into the current approach. Such a comparison would also empirically verify what happens when learning with the prior versus doing inference with an approximate posterior helps.

* The regularization is motivated from the point of view of sampling the hidden states to be from the exponential family, but all the experiments provided seem to use a Gaussian distribution. This paper would be strengthened by a discussion and experimentation with other kinds of distributions in the exponential family.

---

### Official Review · AnonReviewer2 · 2017-11-27
**Running an RNN for one step from noisy hidden states is a valid regularizer**

**Rating:** 3
**Confidence:** 3

**Review:**

In order to regularize RNNs, the paper suggests to inject noise into hidden units. More specifically, the suggested technique resembles optimizing the expected log likelihood under the hidden states prior, a lower bound to the data log-likelihood.

The described approach seems to be simple. Yet, several details are unclear, or only available implicitly. For example, on page 5, the Monte Carlo estimation of Lt is given (please use equation number on every equation). What is missing here are some details on how to compute the gradient for U and Wl. A least zt is sampled from zt-1, so some form of e.g. reparameterization has to happen for gradient computation? Are all distributions from the exponential family amendable to this type of reparamterization? With respect to the Exp. Fam.: During all experiments, only Gaussians are used? why cover this whole class of distributions? Experiments seem to be too small: After all the paper is about regularization, why are there no truely large models, e.g. like state-of-the-art instances? What is the procedure at test time?

---

### Public Comment · ~Aaron_Jaech1 · 2017-10-28
**perplexity on penn tree bank dataset**

I suspect that something may be wrong with the way that you are training your language model. Is it possible that you are overfitting? You don't say the dimensionality of the RNN you used. Knowing that would help interpret your numbers better.

The perplexities in Table 2 seem to get worse when you add more layers. Also, in general these numbers are way higher than expected. The perplexity from your RNN baseline is higher than a 5-gram LM. Your NOSIN model is much worse than Mikolov's 2011 RNN and the state-of-the-art has improved a lot since then.

---

> ### Author Response · Authors · 2017-11-21
> **perplexity on penn tree bank dataset**
>
> Hi Aaron,
>
> Thank you for your questions and remarks. This paper proposes a method for reducing overfitting in RNNs. This consists in injecting noise judiciously into the hidden units of the RNN. We perform training by integrating over the injected noise. This has an ensembling effect to it just like dropout. Our purpose was to compare to the non-regularized RNN and to variational dropout as implemented in Gal et al. 2016 (https://arxiv.org/abs/1512.05287).
>
> In table 2 we used 200 hidden units so it is expected that we don't match the state-of-the-art numbers which correspond to way bigger networks. Again, the purpose was to compare performance of the three types of networks: deterministic, regularized with dropout, regularized with noisin. The benefits we are seeing with our method will be even better with bigger networks where regularization is beneficial. We are adding those results in the revision.

---

### Public Comment · (anonymous) · 2017-11-28
**How is the proposed method compared with the scheme to inject weight noise in RNNs?**

This paper proposed to inject noise in the hidden units of the RNN and then maximize the original RNN’s likelihood averaged over the injected noise.

It seem closely related to [*], which injects noise on the weight of the RNN during training (enhancing exploration of the model-parameter space) and model averaging when testing. [*] also performs regularization, as it yields a principled Bayesian learning algorithm. It is curious to see the performance comparison.

[*] Scalable Bayesian Learning of Recurrent Neural Networks for Language Modeling, ACL 2017

---

### Public Comment · (anonymous) · 2017-12-02
**Reproducibility Challenge**

Greetings to the authors of this paper,

Your paper is very interesting and insightful. As part of a reproducibility challenge (http://www.cs.mcgill.ca/~jpineau/ICLR2018-ReproducibilityChallenge.html) , our team of students would like to attempt at reproducing the results of your paper. We are not affiliated with the official reviewers.

If it would be possible, it would be incredibly helpful if you are interested in providing parts of the code used in your implementations.

If you are interested, please comment below, and we can arrange to contact each other in private.

Thank you

---

### Decision · Program_Chairs · 2018-01-29
**ICLR 2018 Conference Acceptance Decision**

**Decision:**

Reject

**Comment:**

This paper proposes a regularizer for recurrent neural networks, based on injecting random noise into the hidden unit activations.  In general the reviewers thought that the paper was well written and easy to understand.  However, the major concern among the reviewers was a lack of empirical evidence that the method works consistently.  Essentially, the reviewers were not compelled by the presented experiments and demanded more rigorous empirical validation of the approach.

Pros:
- Well written and easy to follow
- An interesting idea
- Regularizing RNNs is an interesting and active area of research in the community

Cons:
- The experiments are not compelling and are questioned by all the reviewers
- The writing does not cite relevant related work
- The work seems underexplored (empirically and methodologically)